# Bond-Slip Performances of Ultra-High Performance Concrete Steel Tube Columns Made of a Large-Diameter Steel Tube with Internally Welded Steel Bars

**DOI:** 10.3390/ma16103836

**Published:** 2023-05-19

**Authors:** Nianchun Deng, He Zhao, Dunrong Yao, Zhiyu Tang

**Affiliations:** 1College of Civil Engineering and Architecture, Guangxi University, Nanning 530004, China; 2010302102@st.gxu.edu.cn (H.Z.); 2010302089@st.gxu.edu.cn (D.Y.); 2Guangxi Key Laboratory of Disaster Prevention and Engineering Safety, Nanning 530004, China

**Keywords:** ultra-high-performance concrete-filled steel tube column, large-diameter, internal welded circular steel bars, bond-slip performance, push-out tests

## Abstract

Large-diameter concrete-filled steel tube (CFST) members are being increasingly utilised owing to their ability to carry larger loads and resist bending. Upon incorporating ultra-high-performance concrete (UHPC) into steel tubes, the resulting composite structures are lighter in weight and much stronger than conventional CFSTs. The interfacial bond between the steel tube and UHPC is crucial for the two materials to effectively work together. This study aimed to investigate the bond-slip performance of large-diameter UHPC steel tube columns and the effect of internally welded steel bars in steel tubes on the interfacial bond-slip performance between the steel tubes and UHPC. Five large-diameter UHPC-filled steel tube columns (UHPC-FSTCs) were fabricated. The interiors of the steel tubes were welded to steel rings, spiral bars, and other structures and filled with UHPC. The effects of different construction measures on the interfacial bond-slip performance of UHPC-FSTCs were analysed through push-out tests, and a method for calculating the ultimate shear bearing capacities of the interfaces between steel tubes containing welded steel bars and UHPC was proposed. The force damage to UHPC-FSTCs was simulated by establishing a finite element model using ABAQUS. The results indicate that the use of welded steel bars in steel tubes can considerably improve the bond strength and energy dissipation capacity of the UHPC–FSTC interface. R2 exhibited the most effective constructional measures, resulting in a significant increase in ultimate shear bearing capacity by a factor of approximately 50 and energy dissipation capacity by a factor of approximately 30 compared to R0 without any constructional measures. The load-slip curve and ultimate bond strength obtained from finite element analysis and the interface ultimate shear bearing capacities of the UHPC-FSTCs obtained using the calculation method agreed well with the test results. Our results provide a reference for future research on the mechanical properties of UHPC-FSTCs and their engineering applications.

## 1. Introduction

In engineering, concrete-filled steel tube (CFST) structures can effectively utilise the advantages of both steel and concrete materials. The lateral constraint offered by the steel tubes can improve the compressive strength, ductility, and energy consumption capacity of the core concrete, whereas the concrete delays the local buckling of the steel tubes [1,2]. Thus, the bearing capacity of the CFST column is significantly higher than the sum of those of the individual steel tube and concrete. Additionally, as permanent templates, steel tubes can reduce the transportation and column assembly costs of the project. Therefore, owing to their high bearing capacity, good ductility, reduced need for manpower, and cost-effectiveness, CFSTs are widely employed in engineering construction. Particularly in super projects, such as long-span arch bridges, other cross-sea bridges, and large deep-water port wharves, large-diameter CFST members have been increasingly used because of their very good mechanical properties, small soil discharge, and relatively simple construction technology [3,4,5,6]. However, as the cross-sections of the members increase, their material consumption and self-weight increase significantly, posing new challenges in engineering construction. Therefore, researchers are seeking new materials with superior mechanical properties.

With the development of concrete technology and the increasing demand for engineering structural performance, the application of ultra-high-performance concrete (UHPC) has become a major research focus. Unlike ordinary concrete, UHPC has ultr-ahigh strength, high fracture toughness, and reliable durability [7,8,9,10,11]. However, UHPC has not been widely used because of its poor relative ductility, which makes it prone to brittle failure during compression. Injecting UHPC into steel tubes to form steel tube UHPC composite structures can effectively improve the ductility of UHPC owing to the constraints provided by the steel tubes [12]. Additionally, this reduces the weight and strengthens the structure compared to CFST [13,14,15]. Studies on the relationship between the strength increase and ductility indices of UHPC-filled steel tube columns (UHPC-FSTCs) and the diameter-to-thickness ratios of the steel tubes [16,17] have found that the two materials are compatible. However, the ductility of UHPC-FSTCs is lower than that of CFSTs, and the strengthening effect of steel tubes on UHPC is insignificant [18]. The use of steel fibers in UHPC-filled steel tube (UHPC-FST) members can significantly improve the ductility, as well as compressive and flexural ultimate bearing capacity, of these members [19,20,21,22,23]. Additionally, the interfacial bond-slip of the steel tubes and core concrete is now being investigated [24,25,26,27]. Researchers have found that the diameter-to-thickness ratio of the UHPC-FST specimens had the greatest impact on the interfacial bond performance of the specimens, and an increase in the steel fiber content improved the interfacial bond strength of UHPC-FSTCs. The bond-slip performance of UHPC restrained by a round steel tube [28,29], square steel tubes [30,31,32], and ultra-high-strength steel tubes [33], have been investigated; a calculation method for obtaining the interfacial bond strength has also been proposed. Although the research interest in UHPC-FST members is increasing, the results reported thus far and the calculation method for obtaining interfacial bond strength is insufficient.

In addition, for certain CFST members, structural measures are often installed on the inner wall of the steel tube to increase the bond strength between the two materials. Shakir-Khalil [34] conducted push-out tests on concrete-filled steel tubular specimens using various types of shear connectors. His results showed that the addition of shear connectors or welded supports produces extrusion and tectonic effects, which can effectively improve the interfacial bond strength. Roeder et al. [35] found that in some cases, the natural bond strength between the steel tube and concrete can ensure the transfer of the axial load, whereas, in others, it is necessary to ensure the transfer of the load through shear connectors. Kaicheng et al. [36] studied the interfaces of concrete-filled steel tubular stub column specimens using different methods. Their study indicated that the rougher the inner wall of the steel tube, the better the bond property. Longitudinal ribs welded on the inner wall of the steel tube significantly improved the interfacial bond strength of the members, which was found to be proportional to the length of the ribs. The number of studs also affected the bond strength. Lihua et al. [37] proposed a new type of concrete-filled steel tube structure that uses raised patterns on the inner wall of a steel tube to improve the natural bond strength between the steel tube and concrete. TAO Z et al. [38] demonstrated through a series of push-out tests that welding inner rings on the inner surface of a steel tube is the most effective method for improving bond strength, followed by welding shear stress nails and using expansive concrete. Vidya et al. [39] conducted pull-out tests to investigate the effects of rebar length, concrete cover, rebar size, and rebar type on bond strength. Their results showed that the extension length of embedded rebar in UHPC can be significantly reduced compared with that of ordinary concrete. Dong Hongying [40] found that studs or steel tie structures can significantly improve the interfacial bond strength and energy–dissipation capacity of UHPC-filled circular steel tubes.

Large-diameter concrete-filled steel tube (CFST) members are utilised more frequently owing to their ability to carry larger loads and resist bending. The outer diameters of the CFST specimens in the existing studies were approximately 200 mm. Because of the small size and structure of the specimens, the actual mechanical properties of CFST members with increasing diameters in engineering applications cannot be accurately reflected [3]. Incorporating UHPC into steel tubes to create UHPC steel tube composite structures produced structures that were lighter in weight and significantly strengthened. However, the interfacial bond between the steel tube and UHPC is crucial for the two materials to work together reliably, and the bond-slip properties of large-diameter steel tube UHPC columns have not yet been reported. Additionally, research on methods to improve the interface bond-slip performance of UHPC-FSTCs is still in its initial stages, and only a few studies exist on the influence of the internal welded steel bar structures on bond performance. Therefore, to investigate the bond-slip performance of large-diameter UHPC steel tube columns and the effect of internally welded steel bars in steel tubes on the interfacial bond-slip performance between the steel tubes and UHPC, this study proposes the welding of a steel bar structure on the inner wall of a steel tube to improve the bearing capacity of the interface of a large-diameter UHPC-FSTC. Through a push-out test, analysis, and calculations using five specimen groups containing different types of UHPC-FSTCs, the influence of the internal welded steel bar structure on the bond-slip performance of the UHPC-FSTC interface was studied. This study provides a reference for future research and engineering applications in the field. Our results provide a reference for future research on the mechanical properties of UHPC-FSTCs and their engineering applications.

## 2. Specimen Design and Test Scheme

### 2.1. Test Specimen Design

Five UHPC-FSTCs were designed and fabricated, and push-out tests were conducted. The specimen parameters are listed in Table 1. Figure 1 and Figure 2 display the design drawings and photographs, respectively, of the specimens and the position of the welded steel bar inside the steel tube. The maximum radial diameter of the steel tube was 370 mm with a thickness of 10 mm and a height of 550 mm. A space of 50 mm was reserved at the bottom of the steel tube where UHPC was absent, and the strength grade of the steel tube material was Q235. The diameter of the steel bars welded inside the steel tube was 10 mm and their strength grade was HPB300. The mechanical properties of the material using the Chinese code GB/T 228.1–2010 [41] are listed in Table 2.

The UHPC premix SBT-UDC (II) purchased from Sobute New Materials Co., Ltd. (Nanjing, China) was used as the core concrete, and the concrete mixing ratios are listed in Table 3. The premix was composed of cement, silica fume, fly ash, and quartz sand in a ratio of 1:0.194:0.258:0.403. The river sand with a fineness modulus of 2.36 and a maximum particle size of 2.36. The strength of the UHPC was obtained by testing standard concrete cubes according to Chinese standards [42]. The average compressive strength and elastic modulus of the resulting concrete cube are 132.05 and 51,081.46 MPa respectively.

### 2.2. Experimental Program

#### 2.2.1. Loading Scheme

Tests were conducted in the experimental area of a 1000-ton hydraulic pressure testing machine in the Hall of Structures of Guangxi University. The loading scheme applied in the test is shown in Figure 3. The specimen being tested was placed with the air-gap end facing down on the bottom steel block and the other end facing up against the top steel block. The diameter of the top steel block was slightly smaller than that of the specimen, and a loading pressure was applied from the top end to the bottom end of the block. The end in contact with the top steel block and that with the air gap is called the loaded and free ends, respectively.

Before the test, a pressure of approximately 15 kN was applied to compact the top end, specimen, and bottom end, thereby eliminating the impact of virtual displacement and adjusting the flat. The loading force was controlled by the displacement, and the loading rate was maintained at 0.5 mm/min. The loading process was terminated when the displacement of the loaded end reached 40 mm.

#### 2.2.2. Layout of Measurement Sensors

The arrangement of the displacement sensors is illustrated in Figure 3. Four displacement sensors were provided with two linear variable differential transformers (LVDTs) arranged at the loaded end to measure the displacement at the top end, and two rotary variable differential transformers (RVDTs) were arranged at the free end. RVDTs convert changes in angular displacement into linear displacements by means of a winding wheel, rope, and a rope encoder. A wooden block was glued to the bottom surface of the UHPC in the reserved gap of the specimen, and the bottom end of the wooden block was connected to a wooden strip. The wooden strip was synchronously displaced with the internal UHPC through its connection to the wooden block. The RVDTs measured the downward displacement of the wooden strip by the drawstring encoders, thereby determining the relative slip between the UHPC at the free end and the steel tube. One end of the wooden block was bonded to the UHPC, and the other end was suspended along a wooden strip. The wooden block could be used for rigid-body displacement with UHPC, preventing excessive deformation of the block and ensuring accurate free end displacement measurement.

Twelve strain gauges were welded to the inner and outer walls of the steel tube to measure the strain at the corresponding points. Strain gauges with lengths of approximately 3 cm were attached to a welded substrate. Fiber grating sensors were welded at three positions along the outer wall of the steel tube, and the arrangement of the strain measurement points is shown in Figure 4. A DH3819 wireless static strain testing system and SA-1102 grating demodulator were used to collect the measurement data.

## 3. Test Results and Analysis

### 3.1. Test Phenomena and Failure Patterns

During the push-out test, the specimens produced a mild creaking sound, which was possibly because of continuous damage to the surface of the UHPC core; the external damage of the specimen was not obvious. The test process for all five specimens followed similar rules. At the beginning of the loading stage, local slip occurred at the loaded and free ends. As the load increased, the slip appeared, and then, the slip area gradually extended to the middle of the specimen, and the sound of brittle fracture was heard. The slip increased linearly with the load. When the sliding speed increased substantially, the load reached a maximum value and then rapidly decreased. At this time, a loud sound was constantly heard, which could be inferred as the sound generated by either the squeezing of concrete, the steel bars welded on the steel tube cutting into concrete, or the welded steel bars breaking. Subsequently, the load continued to increase, the sliding areas at both ends gradually connected to the middle, and the middle area began to slip. Most of the interfaces were in the slip state, and the load-rising rate decreased. As the load increased, the slip increased rapidly until the concrete was moved to the bottom of the steel tube and the test was completed.

Because of the compression of the steel block, the concrete at the edge of the loaded end performed a shear action on the surrounding contact surface, causing the surroundings of the core concrete to be damaged to different degrees. After loading, traces left by friction with concrete were observed on the inner surface of the steel tube at the loading end. By observing the surfaces of each specimen, the concrete near the edge of the loading end was found to be damaged. The interface of the specimen without such a structure was relatively smooth with only a sliding trace. In comparison, the specimen with the steel bar structure had more small concrete fragments. Figure 5 shows the overall failure of the specimens.

### 3.2. Main Results of the Test

By considering the average measurements of the two LVDTs and RVDTs as the slips at the loading end (*S*_l_) and free end (*S*_f_), respectively, the load-slip (*P-S*) curve of each specimen is plotted, as shown in Figure 6. The *P-S* curves corresponding to both ends of each specimen are observed to be consistent, with the curve corresponding to the loading end slightly ahead of that corresponding to the free end. This indicated that the development of the bond forces at both ends of the specimen was consistent. However, because the concrete and steel tubes were stressed at both ends of the specimen, the elastic modulus of the concrete was smaller than that of the steel tube; therefore, the concrete deformed faster than the steel tube, and its slip speed was faster after being stressed.

The *P-S* curves can be divided into three stages (ascending, descending, and residual), where the descending stage indicates a decrease in the interfacial bond properties. To describe the *P-S* curve in detail, three characteristic points were identified on the curve corresponding to the loaded end. These are the starting point (*S*_s_, *P*_s_), where the rapid development of the slip begins, the ultimate point (*S*_u_, *P*_u_), which is the position of the maximum load during slip, and the residual point (*S*_r_, *P*_r_), which is the endpoint of the descending section after the maximum value. Furthermore, Δ_1_, Δ_2_, and Δ_3_ are the differences in the slip values between the loaded and free ends corresponding to the three points; α_1_ is the ratio of the values at *P*_s_ and *P*_u_, and α_2_ is the ratio of the values at *P*_r_ and *P*_u_. The characteristic values obtained from the *P-S* curves are listed in Table 4.

Figure 6 shows that load slip increased proportionally during the ascending stage of the curve. In the descending stage, the curve increased nonlinearly owing to the mutual extrusion force between the UHPC and the steel tube. With an increase in axial deformation, the effect of the UHPC is enhanced. When the load reaches its peak, the curve decreases slightly and rapidly, followed by a gentle upward trend.

Average bond strength *τ* is the ratio of *P* to *A* (the contact area between the steel tube and concrete), and *τ*_s_, *τ*_u_, and *τ*_r_ are the average bond strengths corresponding to *P*_s_, *P*_u_, and *P*_r_, respectively (Table 5). Evidently, the bonding strengths of R2 and R3 are not much different, and the bonding strengths of R1 and RS are also almost the same.

Subsequently, the energy-dissipation capacity of the specimen during the loading process was evaluated. The energy absorbed per unit volume when the displacement of the specimen reached a certain value was defined as *W*, which was calculated as the area under the load-slip curve. The corresponding equation [43] is as follows:(1)$$W=\left(\int_{0}^{S}P\left(S\right)dS\right)/V.$$
where *V* is the specimen volume, *S* is an arbitrary displacement, and load *P* is a function of *S*.

The energy dissipation capacities of the specimens at the early loading stage are of significance to engineering; therefore, the energy dissipation values when the displacement of each specimen reaches 25 mm are discussed [44]. The area corresponding to the first 25 mm slip values on the *P-S* curves was considered to be the representative value *W* of the energy dissipation capacities of the specimens during the slip process. Representative energy dissipation capacities *W*_s_, *W*_u_, and *W*_r_ corresponding to the (*S*_s_, *P*_s_), (*S*_u_, *P*_u_), and (*S*_r_, *P*_r_) points of each specimen are shown in Table 6. Evidently, R2 performed better than R3 in all indices, and RS had a stronger energy dissipation capacity than R1 in the residual stage.

### 3.3. Interpretation of the Results

The test results reveal that for specimen R0, which contains no structural measures, the interfacial shear force between the steel tube and UHPC was jointly borne by the chemical adhesion, mechanical interlocking force, and friction resistance at the initial stage of loading. With a gradual increase in the interfacial shear force, the UHPC in the core area produced a slight longitudinal compression. Additionally, the interface with the wall of the steel tube near the loaded end produced dislocations, which were then transferred downward. The interface dislocation reduced the chemical adhesion and the shear force borne by the small concrete wedge shape of the microgroove on the inner wall of the steel tube increased, thereby increasing the mechanical interlocking force. The UHPC in the core zone expanded transversely while being compressed longitudinally and extruded into the inner wall of the steel tube, thereby increasing the friction between the steel tube and UHPC. Additionally, the mechanical interlocking force increased slightly. At this stage, the slip increased essentially linearly with the load. When the free end was displaced, the slip had been transmitted to the free end, that is, the chemical adhesion had disappeared. However, the mechanical interlocking force and friction resistance were insufficient to resist the interface shear; therefore, the bearing capacity reached a peak, followed by a sudden drop and a sudden increase in slip. The reduction in the load and continuous damage to the surface of the UHPC also resulted in a reduction in friction and the mechanical interlocking force. After a certain amount of slip, the lateral expansion of the concrete caused friction, mechanical interlocking, and shear forces to gradually stabilize. At this stage, the bearing capacity increased slowly, and the relative slip increased relatively quickly until the specimen was pushed out.

When the interface chemical adhesion disappeared in a steel tube with an internally welded steel ring structure, the steel rings at the elastic stage gradually resisted the increasing shear force with increasing deformation. The steel bar with high stiffness behaved similarly to a cantilever structure under a uniform load at the initial stage. As the load increased, the welded part of the steel ring and inner wall of the steel tube bore an increasing load. The transformation from a uniform load state to a non-uniform load state resulted in the transfer of the resultant moment of the distributed force to the welded part of the steel ring. Continuous high pressure caused continuous lateral expansion of the core UHPC, which increased the friction resistance produced by the interface. This is beneficial to the mechanical interlocking force and shear resistance of the steel ring. The increase in the resistance caused the upper pressure to increase further, forming a mutually beneficial mechanism for the interface shear resistance. At this stage, the slip increased essentially linearly with the load. With an increase in the load and the production of a sudden brittle sound, the inflection point of the *P-S* curves of specimens R1, R2, and R3 appeared, and the welding point of the first ring of the steel bars closest to the loaded end began to break down. Consequently, the bearing capacity of the steel ring reached a peak, followed by a sudden drop and a sudden increase in slip, and the bearing capacity descending part (*P*_u_*-P*_r_) was the main bearing force of the first ring. Because only one ring of the steel bar was welded in the R1 specimen, after the welding joint of the steel ring was broken, the shear force stopped increasing and stabilised. The bearing capacities of R2 and R3 suddenly decreased and subsequently began to increase smoothly. At this time, the second ring of the steel bars bore the main load. When the displacement of the loaded end reached 30 mm, the bearing capacity of the R3 specimen suddenly decreased again, and the welding joint of the second ring of the steel bar was destroyed. However, because the slip had entered the residual stage, the bearing capacity increased slowly and the relative slip increased relatively quickly until the specimen was pushed out.

The results showed that the bond strength of R1 was higher than that of R0 as the bearing capacity increased 24-fold, and the energy dissipation capacity of R1 increased 15-fold. The bearing capacity and the energy dissipation capacity of R2 both were double that of R1. The bond strength of R3 was similar to that of R2. The bearing capacity of R3 increased by approximately 7%, whereas its energy dissipation capacity reduced by 14%. On balance, R2 has the relatively best constructional measures, with an approximately 50-fold increase in load-bearing capacity and a 30-fold increase in energy dissipation capacity relative to R0 which does not have any construction measure.

For the RS specimen with inner-welded spiral steel bars, the curve of the descending stage (*P*_u_*-P*_r_) did not change considerably because the spiral steel bars were continuous. After reaching the ultimate bearing capacity, the *P-S* curve increased steadily to 20% higher than that of the R1 specimen.

### 3.4. Strain Distribution

The longitudinal strain of the outer wall of the steel tube varied according to position, as shown in Figure 7. In the figure, *h* is the relative height of the measuring point, and the free end corresponds to 0 mm. Additionally, *ε* refers to the strain.

As the position of the measurement sensors changed, the longitudinal pressure on the steel tube increased uniformly, and the strain of each specimen was triangularly distributed. This indicated that the interface effect on the steel tube was uniformly distributed from the top to the bottom, and a good synergistic effect existed between the steel bars.

## 4. Finite Element Simulation

### 4.1. Constitutive Model of the Materials

The starting point for the finite element analysis is the concrete constitutive model. The constrained concrete constitutive model proposed by Linhai [45] is commonly used, and the hoop coefficient *ξ* governs the interaction between the steel tube and core concrete. The corresponding equation is
(2)$$\xi =A_{s}f_{y}/A_{c}f_{c}.$$
where *A_s_* is the cross-sectional area of the steel tube in mm^2^, *f_y_* is the yield strength of steel in MPa, *A_c_* is the cross-sectional area of the concrete, and *f_c_* is the axial compressive strength in MPa of the concrete design value.

The material composition of UHPC and, therefore, its material properties differ from those of normal concrete because of coarse aggregate exclusion. Therefore, the constitutive model of confined normal concrete is not applicable to the numerical analysis of UHPC steel tubular members. Based on the axial compression test of a steel-tube UHPC stub column reported by Qiuwei et al. [46], a constitutive model of steel-tube-constrained UHPC under axial compression was studied. The characteristic parameters of the constrained UHPC were derived from the results of material performance tests, and the resulting equations are as follows:(3)$$\varepsilon_{0}=\left(1+0.896\xi^{1.1}\right)\times \left(837+223\sqrt{f_{c}}\right),$$
(4)$$\sigma_{0}=\left(1+0.809\xi^{1.311}\right)\times f_{c}.$$

Based on the constitutive model proposed by Linhai [45], a modified constitutive model of a UHPC confined by a steel tube under axial compression was established, and the corresponding equations are
(5)$$y=Ax-Bx^{2}\;\;\;\;\;\;,\;\;\;\;\;\;\;\;\;\;\;\;\;\;\;(x\leq 1),$$
(6)$$y=\left\{\begin{array}{c} \;\;\;\;\;\frac{x}{\beta {\left(x-1\right)}^{2}+x}\;\;\;\;\;\;\;\;\;\;\;\;\;,\;\;\;\;\;\;\;\;\;\;\;0<\xi \leq 0.6\\ \left(1-q\right)+\;\;q\;x^{0.2\xi }\;\;\;\;\;\;\;,\;\;\;\;\;\;\;\;\;\xi >0.6 \end{array}\right.\;\;\;\;\;.$$

In Equation (6), *x* = *ε*/*ε*_0_; *y* = *σ*/*σ*_0_; *A* = 2 − *k*; *B* = 1 − *k*; *k* = 0.434*ξ*;

$\beta ={\left(2.36\times 10^{-5}\right)}^{\left[0.25+{\left(\xi -0.5\right)}^{7}\right]}\cdot f_{c}^{2}\cdot 3.51\times 10^{-4}$, $q=0.434\xi /\left(22.957\xi -9.081\right)$.

The ideal elastoplastic model was used for both the steel tube and steel bars, and the model is expressed as follows:(7)$$\left\{\begin{array}{l} \sigma_{s}=\\ \sigma_{s}= \end{array}\right.\begin{array}{c} E_{s}\varepsilon_{s}\\ f_{y} \end{array}\begin{array}{cc} , & \varepsilon_{s}\leq \varepsilon_{y}\\ , & \varepsilon_{s}\geq \varepsilon_{y} \end{array}$$

The yield strength *f_y_* and elastic modulus *E_s_* of the steel components are listed in Table 3. Poisson’s ratio *ν_s_* of the steel components in the elastic stage was set to 0.3.

### 4.2. Boundary Conditions and Meshing

At the loaded end of the specimen, the UHPC surface was subjected to a unidirectional axial load using displacement control. The steel tube was simulated by using a 4-node reduced integral shell element (S4R). To satisfy the calculation accuracy requirements, a Simpson integral with nine integration points was used in the thickness direction of the shell element. The steel bars were simulated using a linear truss element (T3D2). A three-dimensional solid element in an 8-node reduced integral format (C3D8R) was used to represent concrete. To reduce simulation time, the calculation samples were adjusted, the calculation accuracy was ensured, and the seed distribution and mesh division suitable for the model were calculated. The mesh was divided into 10-mm elements. The structured grid division technology was used to divide the model into units, as shown in Figure 8.

### 4.3. Modelling of the Interfaces between the Steel Tube and UHPC

The interfacial bond and damage evolution between the steel tube and UHPC were simulated using cohesive contact, that is, surface-to-surface contact. The master surface was chosen as the inner wall of the steel tube, the nodes on the UHPC bond surface were selected as the slave nodes, and the contact property was cohesive behaviour. The cohesive model [47] uses the traction–separation criterion to describe the relationship between interfacial cohesion and relative displacement. To define the cohesive zone model in ABAQUS, three-stage parameters must be determined, that is, the slope or stiffness of the linear elastic stage, damage initiation criterion, and damage evolution process. The most common bilinear cohesion model is shown in Figure 9, which is divided into linear elastic behaviour before damage and linear softening behaviour after damage with the damage onset point as the boundary.

The initial elastic behaviour of the cohesion model is expressed by an elastic constitutive matrix linked to the interfacial cohesion and relative displacement of the interface, as expressed in Equation (8).
(8)$$t=\left\{\begin{array}{l} t_{n}\\ t_{s}\\ t_{t} \end{array}\right\}=\left\{\begin{array}{ccc} K_{nn} & 0 & 0\\ 0 & K_{ss} & 0\\ 0 & 0 & K_{tt} \end{array}\right\}\left\{\begin{array}{l} \delta_{n}\\ \delta_{s}\\ \delta_{t} \end{array}\right\},$$
where *t_n_*, *t_s_*, and *t_t_* are the traction stresses acting on the normal and two tangential axes, respectively, of the interface; *δ_n_*, *δ_s_*, and *δ_t_* represent the relative displacements in the normal and two tangential directions, respectively; *K_nn_*, *K_ss_*, and *K_tt_* are the corresponding elastic stiffnesses, which can be calculated as *K = t*^0/^*δ*^0^, where *t*^0^ and *δ*^0^ are the initial stress and initial relative displacement of the interface damage, respectively. This study used the decoupled form, that is, entering *K_nn_*, *K_ss_*, and *K_tt_* and three constant values as the parameters [48].

During the axial push-out process, no opening or tear damage occurred in the circumferential constraint, and only slip damage occurred along the buried depth. The maximum nominal stress criterion was used as the damage-initiation criterion.
(9)$$\frac{t_{s}}{t_{s}^{0}}\geq 1,$$
where $t_{s}^{0}$ denotes the critical stress of the tangential (slip-type) damage. When the stress ratio reached 1, that is, the adhesive stress reached *τ*_0_, damage began to occur. The interaction between steel and concrete was simulated by using embedded elements.

### 4.4. Comparative Analysis of Interfacial Bond Strength

Based on the finite element modelling process above, five cohesive force models of the steel-tube UHPC specimens were established. The cohesive force nephograms of the inner walls of the steel tubes of the five specimens when the displacement of the loaded end was 10 mm are displayed in Figure 10.

Figure 10 reveals that when the displacement of the loaded end reached 10 mm, the bond stress on the inner wall of the steel tube of R0 was at a minimum, and the bond stresses on the steel tubes of R1, R2, and R3 were at a maximum at both ends of the steel tube and extended gradually to the middle of the steel tube. With an increase in the number of welded steel rings, the bonding stresses on the inner walls of the steel tubes decreased because the welded steel rings contributed to the interface bearing capacity. The bond stress of the RS specimen was close to that of the R1 specimen, which is consistent with the experimental results. Additionally, the bond stress of the RS specimen was the largest at the free end because of the existence of a spiral bar, and the distribution of the bending stress in the steel tube improved.

The above analysis indicates that a steel tube with an internally welded steel bar structure can effectively improve the interface bearing capacity of UHPC-FSTCs and that the welded spiral steel bar structure can improve the interface stress distribution.

### 4.5. Validation of the Finite Element Model

According to the solution obtained from and the post-processing conducted by the software, the corresponding *P-S* curve of each specimen can be plotted, as shown in Figure 11, which compares the simulation and test results. The test and finite element simulation results for the peak loads of each specimen are shown in Table 7.

Figure 10 demonstrates that the *P-S* curves obtained from both the tests and simulations of the five specimens share similar characteristics, and the ultimate bearing capacity and its corresponding displacement are in good agreement. Table 7 shows that the average error between the simulated and tested values of the ultimate bearing capacity is 4.91%. Furthermore, the simulation results of the specimens are in good agreement with the test results. The ultimate bearing capacity of the interface of the 5 specimens exceeded the test values. This deviation may be attributed to the inhomogeneity during the process of filling the steel tube with UHPC, as well as the slight cracking of the reinforcement weld during the loading process of the specimen.

The modelling method, material constitutive model, boundary condition, meshing, and contact condition used in this study can effectively be used to simulate the bond-slip behaviour of steel-tube UHPC specimens and can be used for structural analysis and calculations in practical engineering.

## 5. Calculation of Ultimate Bearing Capacity

### 5.1. Equation for Calculating Ultimate Bearing Capacity

The calculation method for obtaining the ultimate bearing capacity of the UHPC–FSTC interface with internally welded steel bars can be explained based on the interface bearing capacity of a concrete-filled steel tubular column. Additionally, the pertinent design code for the bearing capacity of reinforced concrete structures with stirrups can be used. The bond-slip ultimate bearing capacity (*P_u_*) of the UHPC-FSTC with internally welded steel bars was considered to be the superposition of *P*_*u*0_ and the enhanced bearing capacity component of the steel bars in this study. Here, *P*_*u*0_ represents the ultimate bearing capacity of the UHPC-FSTC interface without steel bars. Based on the shape of and characteristic values obtained from the *P-S* curves, a method for calculating the ultimate bearing capacity of UHPC-FSTCs with different structures was proposed.

For the UHPC-FSTC with one steel ring welded internally, *l*_1_ is the depth at which the steel ring of R1 in the UHPC (the distance between the steel ring and UHPC surface at the loaded end), and the interface ultimate bearing capacity *P*_*u*1_ of R1 is
(10)$$P_{u1}=P_{u0}+\frac{A_{r}f_{c}{}^{\prime }}{\frac{l_{1}}{l_{1}}},$$
where *A_r_* is the projection area of the steel ring and *f_c_*^′^ is the compressive strength of the concrete.

For the UHPC-FSTC with two internally welded steel rings, the interface ultimate bearing capacity *P_u_*_2_ of R2 is as follows:(11)$$P_{u2}=P_{u0}+\frac{A_{r}f_{c}{}^{\prime }}{\frac{l_{2a}}{l_{1}}}+\frac{A_{r}f_{c}{}^{\prime }}{\frac{l_{2b}}{l_{1}}},$$
where *l*_2*a*_ and *l*_2*b*_ are the depths at which the first and second steel rings of R2 were placed in the UHPC.

For UHPC-FSTC with three steel rings welded internally, the ultimate bearing capacity of the first steel ring of R2 was considered to be equal to that of the first steel ring of R3. This is because the welding position of the first steel ring of R3 failed during the test. As the slip of the third steel ring when bearing the main load entered the residual stage, the interface bearing capacity of the third steel ring can be ignored. The ultimate interface-bearing capacity *P_u_*_3_ of R3 is expressed as follows:(12)$$P_{u3}=P_{u0}+\frac{A_{r}f_{c}{}^{\prime }}{\frac{l_{2a}}{l_{1}}}+\frac{A_{r}f_{c}{}^{\prime }}{\frac{l_{3b}}{l_{1}}},$$
where *l*_3*b*_ is the depth at which the second steel ring of R3 is placed in the UHPC.

For the UHPC-FSTC with two coils of spiral steel bars welded internally, the interface ultimate bearing capacity *P_uS_* of specimen RS is:(13)$$P_{uS}=P_{u0}+\frac{A_{r}f_{c}{}^{\prime }}{\frac{\sum_{1}^{n}l_{S}}{nl_{1}}}.$$
where *l_S_* is the stirrup spacing of, and n is the number of spiral steel bars in RS.

### 5.2. Comparison of the Calculated and Test Values

To verify the accuracy of the calculation method, test parameters were substituted into the calculation equation to validate the calculated interface-bearing capacities of the specimens. The calculated values were compared with the test value. Further details are presented in Table 8. The comparison showed that the calculated value to the test value ratio was in the range of 0.98–1.05 with an average value of 1.001, a standard deviation of 0.0231, and a coefficient of variance of 0.0229. This indicates that the calculation method agrees with the test conducted in this study.

In summary, the equation for calculating the ultimate bearing capacity presented in this study is accurate for UHPC-FSTCs with internally welded steel bars. Owing to a lack of relevant research, further experiments and numerical simulations are required to verify the applicability of this equation in additional cases.

## 6. Conclusions

In this study, steel bars were welded to the inner walls of large-diameter steel tubes to strengthen the interfacial bearing capacities of UHPC-FSTCs. A push-out test was conducted on the UHPC-FSTC specimens with different internally welded steel-bar constructions, the test results were analysed using parameters, and the impact of the parameters on the bond-slip performance of the interface of the steel-tube UHPC columns was discussed. The test results were verified using finite element analysis, and the following conclusions were drawn:(1)The bond strength and energy dissipation capacity of the interface of a large-diameter UHPC-FSTC can be considerably improved by adding welded steel bars, with the number of steel rings increasing, the bond strength of the interface also increases and the energy dissipation capacity first increases and then decreases.(2)A comparison of the construction measures of the steel-tube UHPC columns in this study shows that construction measure R2 has the best relative performance, with a bond strength increase of approximately 50 times and an energy dissipation capacity increase of approximately 30 times compared to R0 that does not have any construction measures.(3)A finite element model of UHPC-FSTC specimens with internally welded steel bars was established. The simulation results agreed with the test results in terms of the load-displacement curves, failure modes, and ultimate bearing capacities, thereby proving the applicability of the model. The interface bond strength analysis showed that the welded steel rings inside the steel tubes substantially increased the interface-bearing capacities of the UHPC-FSTCs. The welded spiral bar inside the steel tube improved the interfacial stress distribution.(4)The method for determining the ultimate bearing capacities of the interfaces of UHPC–FSTCs with internally welded steel bars presented in this study is highly accurate. The calculated values agree with the test values. These results provide a reference for future research on the mechanical properties of UHPC-FSTCs.(5)Our findings afford valuable insights for advancing engineering applications of UHPC-FST members, including their use in arch rings for large-span arch bridges, piers, foundations of cross-sea bridges, and deep-water pier foundations. Compared with ordinary CFST, UHPC-FST members have smaller cross-sections and lighter weight. Furthermore, internal welding reinforcement structures increase interfacial bond strength, thus mitigating the debonding problem and improving load-bearing capacity and durability.(6)To explore the application prospects of UHPC-FSTCs in marine environments, the effects of salt corrosion and temperature on the interfacial bonding of steel-tube UHPC columns will also be studied.

## Figures and Tables

**Figure 1 materials-16-03836-f001:**
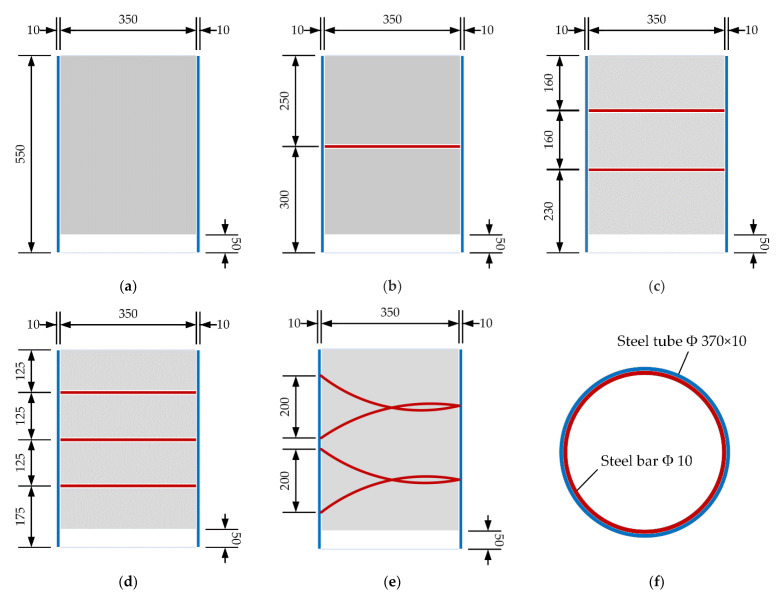
Design drawings depicting the measurements used to construct the specimens (all dimensions are in mm) (**a**) R0; (**b**) R1; (**c**) R2; (**d**) R3; (**e**) RS; (**f**) Top view.

**Figure 2 materials-16-03836-f002:**
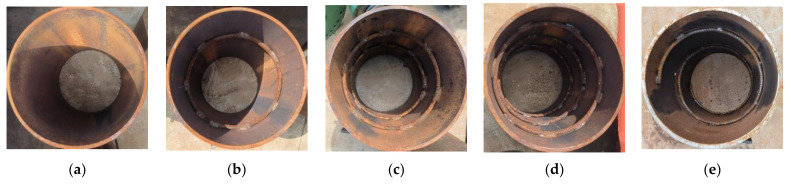
Fabricated UHPC-FSTC specimens (**a**) R0; (**b**) R1; (**c**) R2; (**d**) R3; (**e**) RS.

**Figure 3 materials-16-03836-f003:**
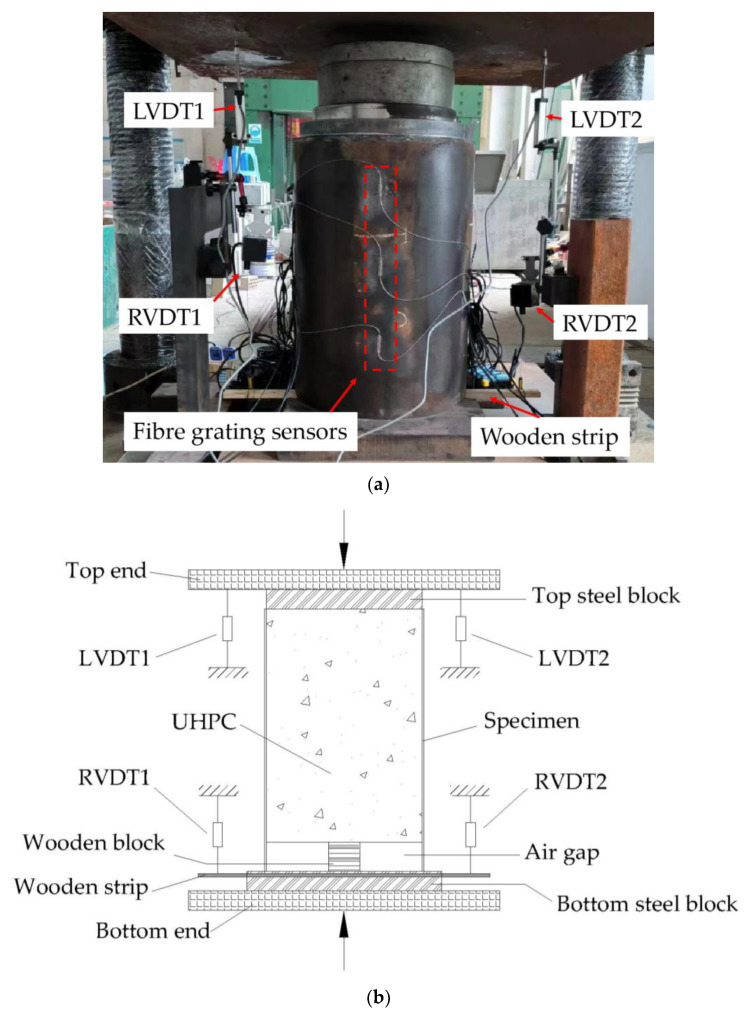
Loading test setup (**a**) Loading test setup; (**b**) Two-dimensional schematic of the loading test setup; (**c**) Three-dimensional schematic of the loading test setup.

**Figure 4 materials-16-03836-f004:**
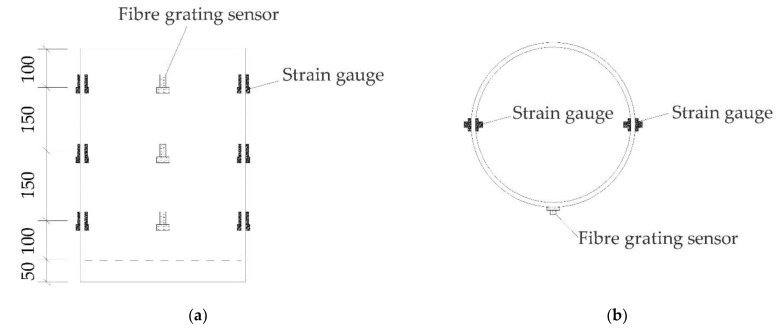
The layout of the measurement sensors (all dimensions are in mm) (**a**) Front view; (**b**) Top view.

**Figure 5 materials-16-03836-f005:**
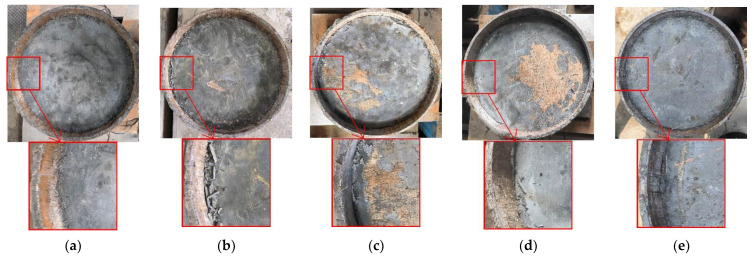
Failure pattern of specimen (**a**) R0; (**b**) R1; (**c**) R2; (**d**) R3; (**e**) R4.

**Figure 6 materials-16-03836-f006:**
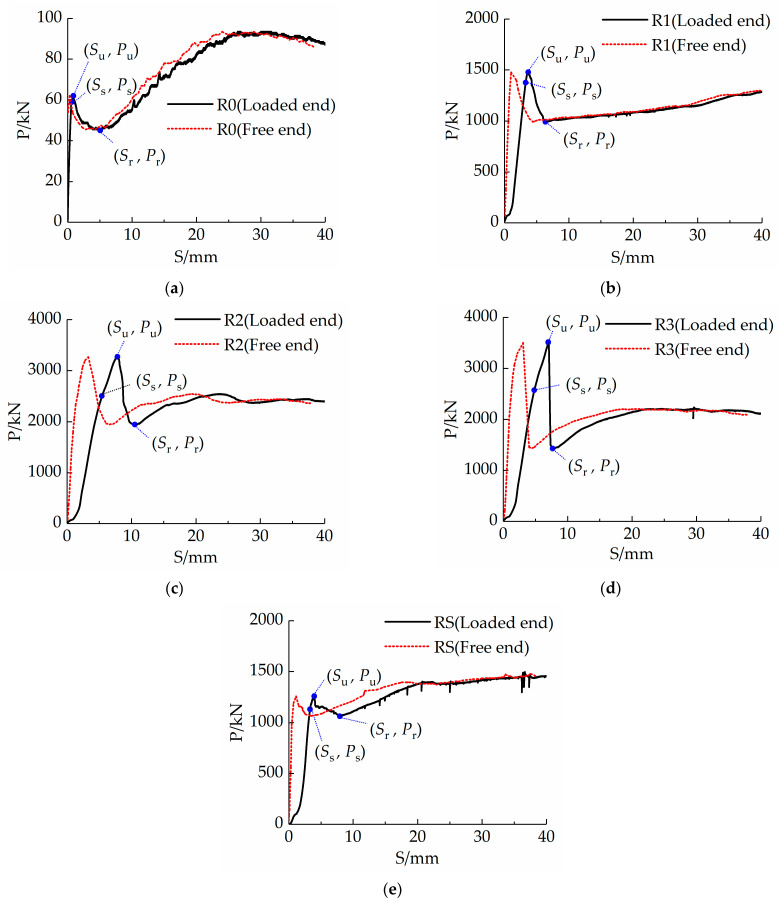
Load-slip (*P*-*S*) curves (**a**) R0; (**b**) R1; (**c**) R2; (**d**) R3; (**e**) RS.

**Figure 7 materials-16-03836-f007:**
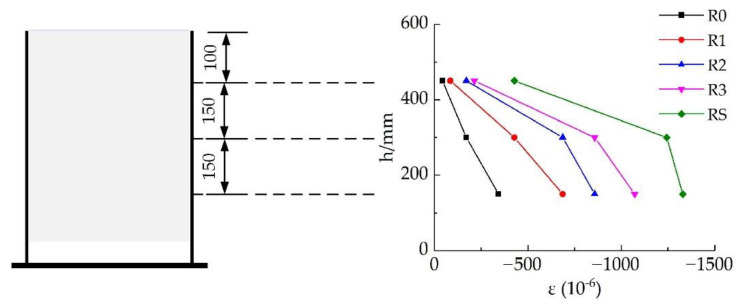
Longitudinal strain on the outer wall of each steel tube specimen (all dimensions are in mm).

**Figure 8 materials-16-03836-f008:**
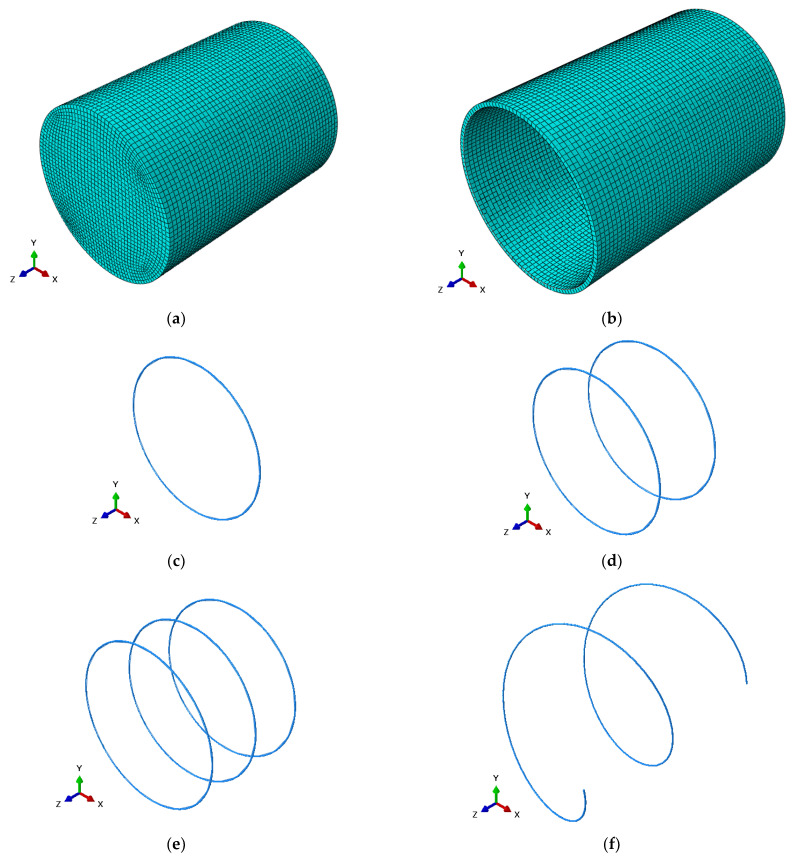
Finite element model of (**a**) Core concrete; (**b**) Steel tube; (**c**) R1 steel bar; (**d**) R2 steel bar; (**e**) R3 steel bar; (**f**) RS steel bar.

**Figure 9 materials-16-03836-f009:**
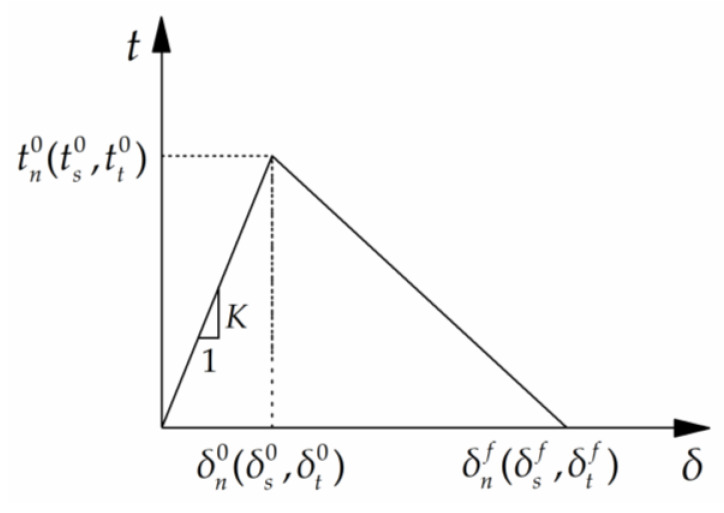
Bilinear cohesion model.

**Figure 10 materials-16-03836-f010:**
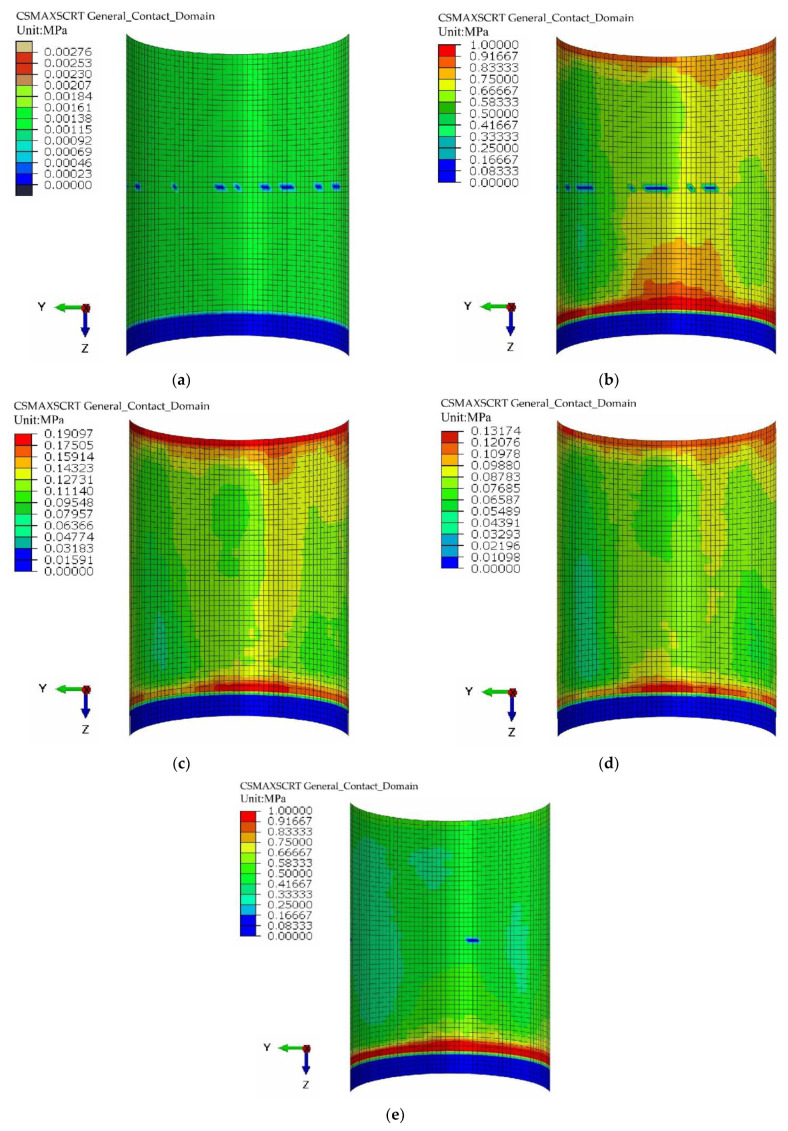
Cohesion of the inner walls of the steel tubes. (**a**) R0; (**b**) R1; (**c**) R2; (**d**) R3; (**e**) RS.

**Figure 11 materials-16-03836-f011:**
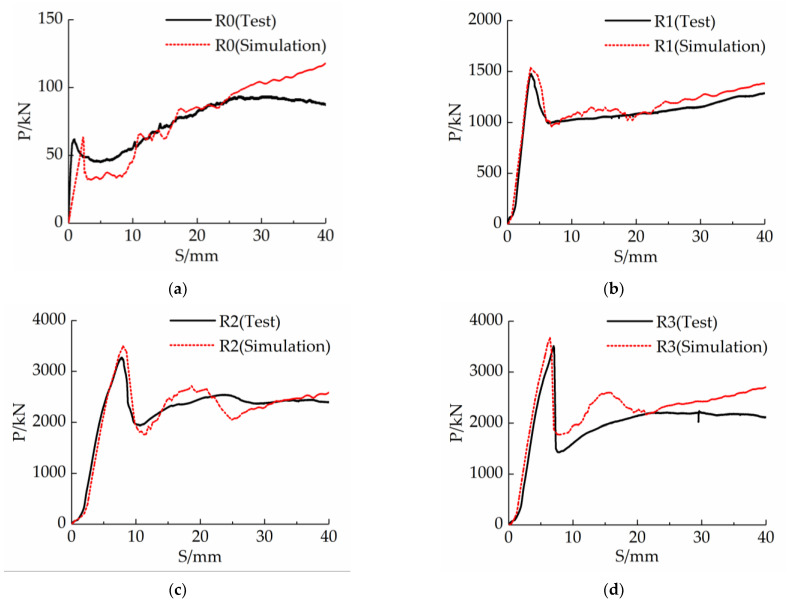
Comparison between the *P*-*S* curves obtained using the simulation and test results (**a**) R0; (**b**) R1; (**c**) R2; (**d**) R3; (**e**) RS.

**Table 1 materials-16-03836-t001:** Specimen parameters.

Specimen ID	Core Concrete	Specimen Size D × t × l (mm)	Interfacial Bond Length (mm)	Number of Steel Rings (pcs)	Number of Spiral Bars (pcs)	Stirrup Spacing (mm)
R0	UHPC	370 × 10 × 550	500	0	0	-
R1	UHPC	370 × 10 × 550	500	1	0	@250
R2	UHPC	370 × 10 × 550	500	2	0	@160
R3	UHPC	370 × 10 × 550	500	3	0	@125
RS	UHPC	370 × 10 × 550	500	0	1	@200

Note: A steel bar structure was not welded inside R0. One, two, and three rounds of steel rings were welded inside R1, R2, and R3, respectively. The interior of RS was welded to two coils of a spiral-steel bar structure. The unit pcs stands for pieces.

**Table 2 materials-16-03836-t002:** Mechanical properties of the steel components.

Size (mm)	Position	*f*_y_ (MPa)	*f*_u_ (MPa)	*E*_s_ (MPa)	*δ* (%)
10	Steel tube	235	380	2.00 × 10^5^	25
10	Steel bar	300	425	1.98 × 10^5^	10

Note: *f*_y_—yield strength, *f*_u_—tensile strength, *E*_s_—the modulus of elasticity, and *δ*—the elongation.

**Table 3 materials-16-03836-t003:** Concrete mixing ratios (kg/m^3^).

Premix	River Sand	Water Reducer	Water	Steel Fibre
1150.8	920.6	13	184.7	181

**Table 4 materials-16-03836-t004:** Characteristic values obtained from the load-slip curves.

Specimen ID	*P*_s_ (kN)	*P*_u_ (kN)	*P*_r_ (kN)	*S*_s_ (mm)	*S*_u_ (mm)	*S*_r_ (mm)	Δ_1_ (mm)	Δ_2_ (mm)	Δ_3_ (mm)	*α* _1_	*α* _2_
R0	58.5	61.5	45	0.58	0.82	5.05	0.52	0.63	0.78	0.95	0.73
R1	1448.5	1475.5	991.5	3.48	3.71	6.45	2.46	2.41	2.06	0.98	0.67
R2	2565.5	3273.5	1940.5	5.59	7.82	10.67	3.96	4.52	4.07	0.78	0.59
R3	2725.7	3512.3	1426.5	5.13	7.01	7.74	3.61	3.93	3.72	0.78	0.41
RS	1186.5	1258.5	1062.5	3.48	3.94	7.91	3.53	2.82	4.43	0.94	0.84

**Table 5 materials-16-03836-t005:** Average bond strengths.

Specimen ID	R0	R1	R2	R3	RS
*τ*_s_ (MPa)	0.11	2.63	4.67	4.96	2.16
*τ*_u_ (MPa)	0.11	2.68	5.95	6.39	2.29
*τ*_r_ (MPa)	0.08	1.80	3.53	2.59	1.93

**Table 6 materials-16-03836-t006:** Energy dissipation capacities.

Specimen ID	R0	R1	R2	R3	RS
*W* (kN∙m)	1.630	25.186	52.915	46.271	28.058
*W*_s_ (kN∙m)	0.024	1.727	5.697	5.157	1.113
*W*_u_ (kN∙m)	0.039	2.322	12.613	11.349	1.896
*W*_r_ (kN∙m)	0.244	5.531	19.077	12.775	6.404

**Table 7 materials-16-03836-t007:** Comparison of simulated and tested values of ultimate bearing capacity.

Specimen ID	Tested Values (kN)	Simulated Values (kN)	Tested Values/Simulated Values	Error (%)
R0	61.5	63.26	0.97	2.78
R1	1475.5	1535.94	0.96	3.94
R2	3273.5	3498.19	0.94	6.42
R3	3512.3	3670.47	0.96	4.31
RS	1258.5	1354.94	0.93	7.12
Average Error				4.91

**Table 8 materials-16-03836-t008:** Calculated and tested values of the ultimate bearing capacities.

Specimen ID	Test Values (kN)	Calculated Values (kN)	Calculated Values/Test Values
R0	61.5	61.50	1.000
R1	1475.5	1471.98	0.998
R2	3273.5	3367.31	1.029
R3	3512.3	3675.86	1.047
RS	1258.5	1236.90	0.983
Average value of the ratio			1.001
Standard deviation of the ratio			0.0231
coefficient of variance of the ratio			0.0229

## Data Availability

The data that support the research can be obtained from the corresponding author upon reasonable request.

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
