# Peer review of "Bond-Slip Performances of Ultra-High Performance Concrete Steel Tube Columns Made of a Large-Diameter Steel Tube with Internally Welded Steel Bars"

_materials, 2023, doi:10.3390/ma16103836_

Round 1

Reviewer 1 Report

Dear authors,

You have done a good research work on "Bond-slip performances of ultra-high performance concrete 2 steel tube columns made of a large-diameter steel tube with internally welded steel bars ". The following things can be considered while revision.

1) Your title and abstract are good.

2) The introduction can be improved with more references and literature reviews.

3) Make sure that the figures and the details of figures come in same page. for example pages 3-4.

4) You can reduce some figures/graphs. In the same figure, you have given 4-5 related figures.

It is good

Reviewer 2 Report

The main observations are listed below. The acceptance of the manuscript would depend on the revision. The author needs to provide a point-by-point response or provide a rebuttal.

1) The abstract should be briefly written to describe the purpose of the research, the principal results, and major findings. Authors should revise it.

2) Authors are encouraged to indicate again the main practical applications of this work.

3) For general readers, authors are encouraged to discuss other kind of works on concrete structures such as: [(a) “Machine learning models for predicting the compressive strength of concrete containing nano silica”, Computers and Concrete, 30(1), 33-42.; (b) “Stability and dynamic analyses of SW-CNT reinforced concrete beam resting on elastic-foundation”, Computers and Concrete, 25(6), 485-495.].

4) Fig. 6 should be more discussed.

5) Conclusion section is poor. Some applications of the model and future scope should be included.

I believe that the above changes will certainly add value to the already well-documented contribution by the authors. With these modifications, I think this already very good article can be improved somewhat and will be better. I then welcome it for publication after revision.

Agree

Reviewer 3 Report

This is an interesting study about the bond-slip performance of UHPC steel tube columns. The paper has the advantage of presenting experimental data as well a finite element analysis. Overall, the paper is well structured, and it is worth to be considered for publication in materials. However, some important issues need to be clarified prior publication.

-Table 1: Does pcs stand for pieces? Also use an asterisk or other symbol at R0 for the footnote below.

- 2.2.2. The authors should provide the nominal measuring ranges of the LVDTs and RVDTs. Also, the length of strain gauges should be given.

- 3.1. The authors should avoid description like a "Ta Da" sound and rather use the term as later in the paper, sound of brittle fracture.

- 3.2. The RVDTs should measure angular displacement. Do the authors back-calculate the displacement/slip from the angular displacement of the wood?

- Figure 6. What is the criterion of defining the (Ss, Ps) point? The authors state that this is the point where the rapid development of the slip begins. Did the authors evaluate the points by looking into the slope of PS curves or the rates of the slip? In any case, it is rather difficult to see a change of the slope on figures (a) and (b) and I would say that for (c) and (d) this point may also be taken at lower/earlier load/time.

- 3.2, page 9, line 207: The authors state in the text that W is the area under the stress-strain curve. However, the equation (1) expresses W in relation to the area of force-displacement/slip curve. Please check and correct accordingly.

- Table 5: I suggest that you also check the W for the three selected points (τs, τu, τr) to see how the individual values compare among the 5 specimens.

- 3.3, page 10, lines 245-246: "As the load increased...bore increasing load." Rephrase the sentence such is comprehensive. 

-3.3, page 10, line 254: inflexion point->inflection point.

- Equation 9. Check if it is >= or <= 1. In the first case what would be the upper limit of this criterion for a fully damaged state?

- Figure 10. Add the units in the colorbar and add also in the caption what is shown in each subfigure (a)-(e).

- 5.1, page 18, line 410 and Equation (10). Is l1 the depth of the steel ring and the second l1 is the distance between the steel ring and the loaded end? Please explain better why you are using the two l1s.

- The FEM model could be potentially used for cases that are not covered by the experiments. Why the authors did not consider such a study which could serve as a basis for further research into the topic?

English is overal fine. 

Reviewer 4 Report

SUMMARY

The article submitted for review is devoted to a topical issue. The bond-slip performances of ultra-high performance concrete steel tube columns made of a large-diameter steel tube with internally welded steel bars are given. The relevance of the study is due to the high efficiency, theoretically and practically proven, of the use of tube concrete structures in modern construction. The authors have taken an interesting approach by combining tube concrete technology with ultra-high performance concrete. Thus, they obtained a number of results verified by finite element methods and test methods, and received confirmation of these results. The authors have done a great job, their research has scientific novelty and practical significance. The study deserves some support, but the article needs serious revision. Below are the comments of the reviewer.

COMMENTS

1.       Authors are encouraged to rework the title of the article. It contains a lot of unnecessary details. In addition, it seems that "bond-slip performances of ultra-high performance concrete steel tube columns" title is too narrow and bulky for a full-fledged study in the journal "Materials". I would like to understand what significant scientific problem the authors solved, what deficit was eliminated in the course of the study. If the authors studied the bond-slip performances of ultra-high performance concrete steel tube columns, then perhaps it should be called that without the word "characteristics". Authors are encouraged to think about the title. In addition, a small technical typo on line 3 in the form of a different font in the word "tube" attracts attention. Authors are encouraged to conduct an editorial proofreading of the article.

2.         The abstract is not quite right. There is no formulation of a scientific problem or a practical problem. The authors immediately proceed to the fact that the influence of internal welded steel bars in steel tubes on the interfacial sliding bond between steel tube and ultra-high performance concrete has been studied. This is not entirely true, because the formulation of the problem must come first.

3.         Methodology, on the contrary, is set out in too much detail. Authors need to work on the abstract structure. At the end, it is reported that the use of welded steel bars in steel tubes can significantly improve the joint strength and energy dissipation of the UHPC-FSTC interface, but these improvements are not quantified. An annotation needs to be worked out.

4.         The introduction is presented in the form of a literature review. At the same time, this literature review does not reflect the current state of the issue of tube concrete structures with high performance concrete inside. The authors need to present more research in the last 5 years because UHPC-FSTC as a materials science and tube concrete structures as a design direction are among the leading ones in modern building scientific practice.

5.         The authors present the methodology in Section 2, but do not describe the materials and methods chosen in great detail. The authors should work on the text part of section 2, because for now it looks like some kind of scientific protocol.

6.         A lot of graphic material is given by the authors, a large number of photographs, diagrams and tables. But all this should be provided with a description and explanations in the analytical part. Otherwise, it will be difficult for the reader to perceive the text of the article.

7.         In addition, the sections in the article are poorly consistent with each other. Clear transitions between sections are necessary. For example, between sections 1 and 2, clear statements of the purpose, tasks of the study, and scientific problem should be added.

8.         Section 2 on the description of methods and materials should be supplemented with a program of test and computational experiments.

9.         I would like to see the presented interesting figure 5 in a higher quality. Here the authors provide photographs of tube concrete filled with a concrete core.

10.      The graphs in Figure 6 are of interest, but they are poorly explained. A more detailed discussion of the results obtained is necessary.

11.      It is necessary to compare the obtained results with the results of other authors. Then the scientific novelty of the research will be more clearly expressed.

12.      The conclusions should be supplemented with a clear statement of the scientific result and reflect the prospects for the development of this study. Where it is recommended to use such tube concrete columns, why to obtain such ultra-high performance, what degree of responsibility these buildings and structures have, where such structures will be used, all this should be reflected in the conclusions. From the point of view of scientific research and prospects, it is necessary to show what research the authors plan to conduct in the future.

13.      The list of references needs some improvement with additions on references for the last 5 years. Some of the references analyzed by the authors are already outdated. Science has come a long way since then. Authors are encouraged to work on the references.

14.      In general, the comment of the reviewer is as follows. The study is interesting, important and relevant, important scientific results have been obtained, and the results may be of interest for practice. Significant revisions and corrections of the reviewer's comments are needed. After corrections, the reviewer would like to see this article again.

General conclusion: major revisions.

Minor language changes required.

Round 2

Reviewer 3 Report

The authors have adressed all the comments and provided the necessary clarifications. No further comments.

Reviewer 4 Report

The authors significantly improved the manuscript by responding to all comments. The article can be published in its current form.